# Chicken Sperm Cryopreservation: Review of Techniques, Freezing Damage, and Freezability Mechanisms

Yunhe Zong [1,†], Yunlei Li [1,†], Yanyan Sun [1], Gamal M. K. Mehaisen [2], Tianxiao Ma [1] and Jilan Chen [1,*]

1   Key Laboratory of Animal (Poultry) Genetics Breeding and Reproduction, Ministry of Agriculture and Rural Affairs, Institute of Animal Science, Chinese Academy of Agricultural Sciences, No.2 Yuanmingyuan West Road, Beijing 100193, China
2   Department of Animal Production, Faculty of Agriculture, Cairo University, Giza 12613, Egypt
*   Correspondence: chenjilan@caas.cn; Tel.: +86-10-6281-6005
†   These authors contributed equally to this work.

**Abstract:** Ex situ preservation is an important method in the preservation of chickens, and cryopreservation of semen is the only method for gamete preservation at present. During the last two decades, many studies have been performed to develop standard chicken semen cryopreservation technology and achieve great progress. Many attempts and methods were investigated to adapt subspecies or different breeds. In this paper, we firstly reviewed the main factors affecting cryopreservation of chicken sperm, including the unique structure and characteristics of the spermatozoa. Secondly, the studies on key points of the chicken sperm cryopreservation technology, including semen dilution, cryoprotectants, equilibration time, packaging types, and freezing and thawing rates were summarized to generate the optimal parameters. Then, the mechanism underlying freezing damage and freezability revealed by recent omics methods relevant to the efficiency of cryopreservation were discussed. This review will provide relevant reference for the future investigation of poultry semen cryopreservation technology.

**Keywords:** chicken; semen; cryopreservation; cryodamage; freezability





## 1. Introduction

Sperm cryopreservation is one of the most important procedures in the biotechnological development of assisted reproduction in fish [1], birds [2,3], and mammals [4]. Especially in cattle breeding, artificial insemination relies almost entirely on the use of frozen semen from superior males in breeding centers [5,6]. This is even more relevant when it comes to preserving semen from endangered animals. The semen freezing technology has been widely used in the semen preservation of mammals such as pigs, cattle and sheep, and the technology has been relatively mature, but it cannot be directly used for reference in chickens, nor can predecessors. Poultry production and the preservation of poultry germplasm resources also have a strong demand for chicken semen cryopreservation technology. Due to the unique physiological characteristics of avian females, it is difficult to preserve the oocytes or embryos. Thus, the cryopreservation of avian sperm plays a substantial role to preserve endangered wild species and genetic diversity in commercial species. It is, meanwhile, an indispensable strategy to improve the genetic materials conservation and biodiversity protection. Cryopreservation is known as an important tool for programs of genetic diversity management and of endangered breeds' conservation that can be used to preserve the genetic diversity of chicken populations, especially vulnerable and important breeds. Some countries have developed methods for the cryopreservation of poultry semen, but the maturity and stability of the technology are poor. Although some methods show relatively good sperm vitality, the preservation time of chicken semen cryopreservation-thawed sperm vitality fertilization rate and other indicators are not ideal [7]. It was in 1949 that the success of preserving the fertility of chicken sperm was reported [8]. Semen

cryopreservation has now been widely described in many domestic bird species, including turkey, duck, goose, and guinea fowl [9]. However, it is not yet adapted to the commercial chicken industry mainly due to the instability of the fertility of post-thaw chicken sperm and the achievements remain linked to many variable results in practice and need continuous research devotion. The sperm-freezing process is complicated, with critical steps, including male selection, semen collection, semen quality examination, semen dilution, adding cryoprotectants (CPAs), packaging, freezing, removing CPAs (if glycerol is used), and post-thaw sperm evaluation [10,11]. There is no standardized methodology for each step of the cryopreservation procedure, which also contributes to differences among studies. In this review, the research outline of the sperm cryopreservation technology in poultry was presented to better understand the cellular and molecular mechanisms involved in sperm cryobiology. Furthermore, the most important impressions and considerations that must be considered during the application of the technologies were also discussed. Gathering this information and these findings inspires researchers exploring novel and creative approaches to crack the challenges of chicken sperm cryopreservation.

## 2. Unique Structure and Characteristics of Chicken Spermatozoa

Much effort has been made in recent years to preserve poultry semen by cryogenic methods. However, mechanisms involved in protecting poultry spermatozoa against freeze–thaw damage are not yet well understood. The unique sperm cell morphology and physiological characteristics might be the intrinsic reasons [3,5]. Structural damage to chicken and turkey sperm after freezing and thawing has been reported to result in reduced motility and fertility [12]. Chicken spermatozoa are more susceptible to damage during the freezing process due to their relatively lower surface-area-to-volume ratio and a thinner tail than some species, such as human and bovine [13]. Spermatozoa of different species have a common structure, which consist of the head and tail. The tail is divided into mid-piece, principal-piece and end-piece (Figure 1). The length of bird spermatozoa is very variable, about 30~300 μm, and sperm head length ranges from 11 to 21 μm. The head of chicken sperm was slightly cylindrically curved, about 12.5 μm. The top of the head has a conical acrosome of about 2.5 μm. The mid-piece is about 4.5 μm, the mitochondrial matrix is dense and homogeneous, and the principal-piece is about 90 μm [14]. The sperm head size influences the volume of water carried by the cell, and cell volume and shape affect the membrane ratio of surface area to cell water volume, which is associated with their ability to respond to osmotic changes and survival [15]. The heads of the human and bull sperm are generally oval and larger than that of chicken sperm [16,17], which may account for their differences in sperm freezability. Compared to other species, chicken sperm have a longer tail, about 90 μm, which means they are more vulnerable to damage during freezing operations, resulting in lower fertility [14,18].

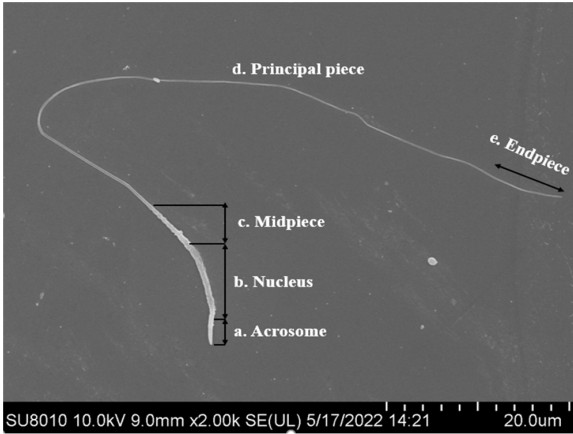

**Figure 1.** The structure of chicken sperm (scanning electron microscope, Hitachi SU8010, Japan. Scale bar = 20 um at 10,000× magnification).

Sperm plasma membrane is considered as an essential factor involved in the resistance of sperm to thermal and osmotic changes during freezing [19]. Bird spermatozoa have little cytoplasmic antioxidants, and the membranes rich in polyunsaturated fatty acids [20]. These characteristics make sperm cells highly susceptible to oxidative stress and increase the production of reactive oxidizing substances, which may lead to reduced motility, DNA damage, and consequently, fertility [21]. Meanwhile, significant damage to the frozen sperm plasma membrane is often associated with the separation of the acrosome, which in turn leads to reduced fertility [22].

## 3. Study on Key Points of the Chicken Sperm Cryopreservation Technology

Current procedures in chicken sperm cryopreservation have been the result of more than 60 years of research. However, no standardized protocol has been developed for all chicken breeds/lines. Many key points, such as semen extenders, CPA, pre-freezing manipulation, semen packaging type, freezing and thawing rates, are all impactful to the cryopreservation efficiency (Figure 2).

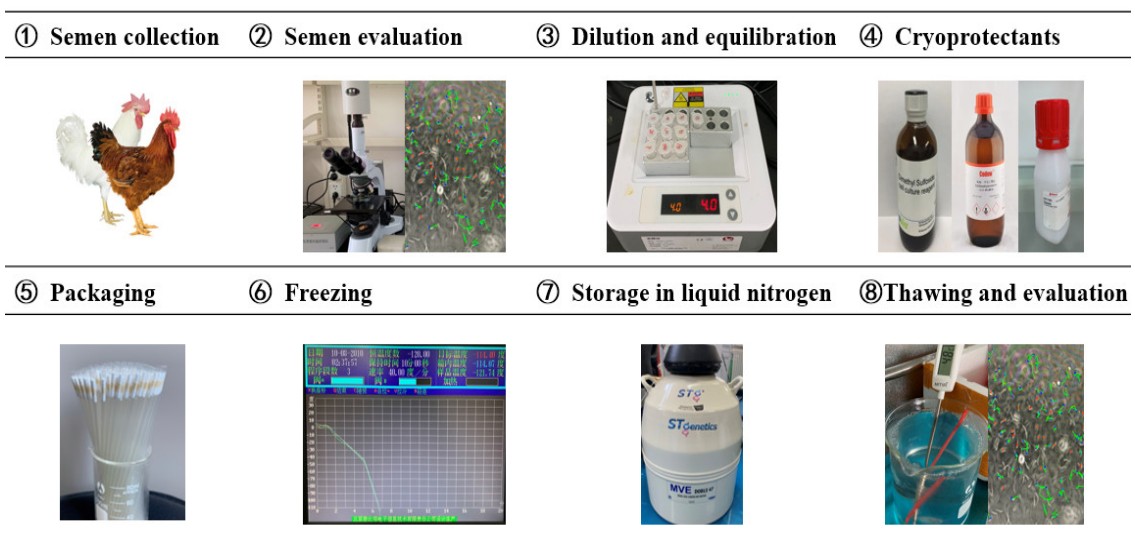

**Figure 2.** Key processes of chicken sperm cryopreservation technology.

### 3.1. Semen Dilution

The ejaculation volume of roosters is 0.2~0.7 mL, and the semen density is about 2~7 billion/mL. Primarily, the ideal extender should be able to provide energy for sperm metabolism and maintain pH and osmolality, which are essential for maintaining cell viability and function [23,24]. Chicken sperm can tolerate a pH range from 6.0 to 8.0 [25], and maintain its fertilizing ability in extenders with osmolality from 250 to 460 mOsm/kg, although the ideal osmotic pressure is 325 to 350 mOsm/kg [26]. Many extenders have currently been used for poultry semen cryopreservation, such as Lake and Ravie (LR) [27], BHSV [28], Beltsville [29], and EK extender [30]. Most extenders are composed of dipotassium phosphate, sodium glutamate, fructose, and sodium acetate in addition to other buffers and salts. However, the subtle differences in reagents and concentrations among extenders will have a dramatic impact on cryopreservation, as an elusive interaction among solutes or solvents exists [31]. There are also some commercially available extenders, including Poultry media® without antibiotics (IMV Technologies, L'Aigle, France), Raptac® (AMP-Lab, GmbH, Münster, Germany) and NeXcell® (IMV Technologies, L'Aigle, France). It was suggested that two extenders (Poultry media® and Raptac®) would be suitable for the cryopreservation procedure, with an applied modification [32]. Dilution is also important in semen freezing and may affect the concentration of sperm during freezing. Dilution as the first step is crucial, which can avoid semen deterioration due to substrate depletion, increased metabolic by-products and condition change [33]. The dilution rate

is dependent on the original semen density, which varies among species and individual. For most reported studies of fish species, the dilution rate of sperm to final volume ranged from 1:2 to 1:10, and the dilution rate of 1:10 was proved to be more proper [34]. The study in endangered mahseer revealed that the motility rates of semen diluted at 1:10, 1:15, and at 1:20 were significantly higher than those at 1:5 dilution [35]. A wide range of dilution rates are used of rams' semen, ranging from 1:1 to 1:16; furthermore, an increase of dilution rate did not affect ram sperm motility but resulted in a significant decrease in the percentage of live normal sperm [36]. Honey bee semen diluent ratios of 6:1 to 12:1 significantly improved sperm post-thaw viability [37]. A dilution rate of 1:1 to 1:4 is usually recommended for chicken semen freezing [12,38], while studies showed no significant differences in fertility [39]. Therefore, the systematic study on the effects of extender parameters on chicken sperm is of great significance for both scientific research and practical production to maintain the viability and fertility of fresh/frozen sperm.

*3.2. Cryoprotectants*

CPAs protect sperm from ice crystal formation and osmotic and chemical stress. The action mode of CPAs is to reduce the amount of ice formation by increasing the total solute concentration. It also regulates the dehydration rate of cells in the freezing process to achieve more gradual dehydration and minimize the possibility of ice formation in cells [5]. Such components can be classified into permeating/intracellular and non-permeating/extracellular subtypes. Intracellular CPAs increase membrane fluidity through the rearrangement of membrane lipids and proteins and partially dehydrate the cell, thereby reducing the formation of intracellular ice crystals [40]. Extracellular CPAs generally form a shield surrounding cells, which can protect cells by reducing extracellular ice crystals [41–43]. Intracellular CPAs include glycerol, dimethylsulphoxide (DMSO), dimethylacetamide (DMA), N-methylacetamide (NMA), ethylene glycol (EG), dimethylformamide (DMF), and some monosaccharides [44–46]. Extracellular CPAs include polyvinylpyrrolidone (PVP), polyethylene glycol (PEG), sucrose, trehalose (disaccharides), and raffinose (trisaccharide) [41–43]. The results of recent studies on chicken semen cryopreservation are summarized in Table 1.

Glycerol is regarded as a good CPA for poultry semen, but its interaction with the females' genital tract will negatively affects sperm motility [47]. The optimal concentration of glycerol and the mechanism of its contraceptive effect have not been fully obtained and elucidated [9]. The glycerol concentration of 4–11% in poultry semen extenders with different effects depending on other constituents of diluents [48]. Although, a very low concentration of 3% was reported [29], the exceeding 8% and 11% were widely used [49]. Further, according to the statistics of the research progresses in the last ten years summarized in this review, 3% and 8% glycerol are used in many studies (Table 2). The researchers compared 2% and 8% glycerol as CPA and obtained the fertility of 34.8% and 45.1%, respectively [50]. The complex centrifugation process during glycerol removal can cause irreversible physical damage to sperm [51]. The 8% and 11% of glycerol were removed by stepwise dilution to a final dilution of 1:4 *v/v*, and the fertility was 28.8% and 2.1%, respectively [52]. Our previous research found that the highest fertility (48.70%) was found for the 5% and 1:2 stepwise dilution combination [53]. Glycerol is the least toxic CPA for sperm, but the addition and later removal of glycerol makes spermatozoa face a rapid change in osmolality from 1300 mOsm/kg (1.0 M glycerol) to 300 mOsm/kg [52]. Osmotic stress may cause substantial damage to spermatozoa, particularly in their membranes [22]. The above results suggested that the glycerol concentration and dilution rate should be screened and optimized simultaneously to find the best combination.

DMA is one of the most effective CPAs for chicken semen cryopreservation [54]. In previous studies, the best fertility rates of 88% and 93% were reported when semen was frozen with 6% DMA [55–57]. In contrast, no changes in chicken sperm viability and motion traits were found between 3% and 6% DMA [58]. The non-contraceptive DMA with satisfying fertility was normally obtained when the semen was cryopreserved as

pellets. Studies have shown that DMA shows good results when sperm is frozen with pellets [51,57]. However, pellet is not suitable for large cryobank programs, and therefore, DMA is not widely used as compared to glycerol. A study showed that when using 6% DMA, a higher fertility rate of 77% was obtained in the gradual in-straw freezing using a controlled liquid nitrogen vapor than with the pellet method, with the fertility of 65% [51]. This indicated the potential of DMA in poultry semen cryopreservation.

DMF was also frequently used as a CPA for chickens and guinea fowl semen cryopreservation [59]. A step freezing programmer was applied for chicken semen cryopreservation with DMF as a CPA in plastic vials and obtained 79% fertility [57]. In recent years, researchers have used 6% DMF for chicken semen cryopreservation and achieved fertility of higher than 90% [60,61]. However, these skills failed in several breeds, including White Leghorn, Rhode Island Reds, and Beijing-You chickens according to our lab data. The reasons for this high fertility are various, perhaps due to differences in extenders or the use of other additives, although they all used the native Thai chickens. The differences between chicken breeds may have a great influence on the fertility of frozen semen. In addition, DMF has been widely used in other poultry species and made some progress. Similarly, a higher post-thawed quality was observed for duck semen frozen with 8% DMF in extender compared to 4%, 6%, and 10% DMF [62]. Interestingly, 6% DMF has no beneficial effects on gander semen during cryopreservation [63].

The study showed that glycerol appeared to induce the greatest osmotic stress, and ethylene glycol caused the least osmotic damage during the subsequent rapid removal of the cryoprotectant [64]. EG improved the motility of chicken spermatozoa after thawing, and especially the addition of ficoll had additional beneficial effects on progressive motility and apoptosis [65]. On the other hand, other studies suggested that EG was not a good CPA for guinea fowl sperm cryopreservation as no fertile eggs were obtained [28].

In addition to intracellular CPAs, the use of extracellular CPAs alone or in combination with intracellular CPAs has been explored. The effect of a combination of monosaccharides (fructose, galactose, glucose, and xylose) and disaccharides (lactose, trehalose, maltose, and sucrose) for semen cryopreservation has been evaluated on various animal species [66]. The combination of glycerol with non-permeating CPAs (egg yolk, fructose, sucrose or trehalose) seems to be the best alternative to reduce glycerol concentration and its contraceptive effects [67]. The combination of DMA and trehalose showed a positive effect on the quality of cryopreserved semen in chickens [68]. Additionally, the cryopreservation of red jungle fowl's semen with 6% PVP has been shown to produce higher fertility than that with glycerol [69].

**Table 1.** Reports on cryopreservation of rooster sperm in recent years.

| CPAs | Breeds | Age/w | Extender | Antioxidant | Packaging | Freezing | Thawing | Thawing Motility/% | Fertility/% | Reference |
|---|---|---|---|---|---|---|---|---|---|---|
| 2% Glycerol | White Leghorn | 30 | Beltsville | NI | 0.25 mL straw | 5 cm above the $LN_2$ for 12 min | 37 °C, 30 s | 43.1 | 49.5 | [70] |
| 3% Glycerol | Ross | 32 | Beltsville | Quercetin | 0.25 mL straw | 4 cm above the $LN_2$ for 7 min | 37 °C, 30 s | 61.6 | 64.2 | [29] |
| 3% Glycerol | NI | 52 | Beltsville | Resveratrol | 0.25 mL straw | 5 cm above the $LN_2$ for 12 min | 37 °C, 30 s | 60.9 | NI | [71] |
| 3% Glycerol | Ross | 30 | Lake | Glutathione | 0.25 mL straw | 4 cm above the $LN_2$ for 7 min | 37 °C, 30 s | 58.5 | 63.8 | [72] |
| 3% Glycerol | Ross | 30 | Lake | CoQ 10 | 0.25 mL straw | 4 cm above the $LN_2$ for 7 min | 37 °C, 30 s | 55.1 | 62.7 | [21] |
| 3% Glycerol | Ross | NI | Nabi | NI | 0.25 mL straw | 4 cm above the $LN_2$ for 7 min | 4 °C, 3 min | 65.4 | 73.1 | [48] |
| 3% Glycerol | Ross | 24 | Beltsville | Hyaluronic Acid | 0.25 mL straw | 4 cm above the $LN_2$ for 7 min | NI | 55.3 | 65.5 | [73] |
| 3.8% Glycerol | Ross | 30 | Lake | Gamma-oryzanol nanoparticles | 0.25 mL straw | 4 cm above the $LN_2$ for 7 min | 37 °C, 30 s | 71.7 | 71.0 | [74] |
| 3.8% Glycerol | Ross | 30 | Beltsville | Crocin Naringenin | 0.25 mL straw | 4 cm above the $LN_2$ for 7 min | 37 °C, 30 s | 74.4 71.2 | 73.1 74.1 | [75] |
| 3.8% Glycerol | Ross | 30 | Beltsville | Quercetin | 0.25 mL straw | 4 cm above the $LN_2$ for 7 min | 37 °C, 30 s | 67.5 | 61.8 | [6] |

**Table 1.** *Cont.*

| CPAs | Breeds | Age/w | Extender | Antioxidant | Packaging | Freezing | Thawing | Thawing Motility/% | Fertility/% | Reference |
|---|---|---|---|---|---|---|---|---|---|---|
| 5% Glycerol | Beijing You chicken | 52 | Lake | NI | 0.5 mL straw | 12 °C/min, 4–44 °C, 40 °C/min, 44–120 °C | 5 °C, 3 min | 37.4 | 48.7 | [53] |
| 5% Glycerol | Ross | 35 | Lake | Xanthine oxidase | 0.25 mL straw | 5 cm above the LN$_2$ for 12 min | 37 °C, 30 s | 61.8 | 60.4 | [76] |
| 6% Glycerol | Black Silkies | 40 | Lake | NI | 0.25 mL straws | 6 cm above LN$_2$ | 5 °C, 3 min | 70.0 | 77.6 | [77] |
| 8% Glycerol | Ross | 30 | Lake | NI | 0.25 mL straw | 4 cm above the LN$_2$ for 7 min | 37 °C, 30 s | 45 | 45.1 | [50] |
| 8% Glycerol | Ross | 28 | Beltsville | NI | 0.25 mL straw | 4 cm above the LN$_2$ for 7 min | 37 °C, 30 s | 70.1 | 60.0 | [78] |
| 8% Glycerol | Ross | 104 | Modified Beltsville | Lycopene | 0.25 mL straw | 4 cm above the LN$_2$ for 7 min | 37 °C, 30 s | 68.1 | 62.2 | [79] |
| 11% Glycerol | D+/D, R+ | 30–40 | Lake | NI | 0.5 mL straw | –7 °C/min | 4 °C, 3 min | NI | 83.3 | [55] |
| 8% EG | Ghagus | 32 | Lake and Ravie | NI | 0.5 mL straw | 4.5 cm above LN$_2$ for 30 min. | 5 °C, 100 s | NI | 48.1 | [45] |
| 6% DMA | Rhode Island Red | 32–36 | LCM | NI | pellet | dropping directly into LN$_2$ | 60 °C | NI | 79.0 | [80] |
| 6% DMA | Black Silkies | 17−18 | Lake and Ravie | NI | 0.25 mL straw | 7 cm above the LN$_2$ for 7 min | 60 °C | NI | 77.6 | [51] |
| 9% DMA | Hi-Line White | 28 | Lake | NI | 0.25 mL straw | 3 cm above the LN$_2$ for 10 min | 38 °C, 30 s | 24.2 | 45.0 | [81] |
| 6% DMF | Rhode Island Red | 52-104 | Schramm | NI | 0.5 mL straw | 11 cm above the LN$_2$ for 12 min, 3 cm above the LN$_2$ for 5 min | 5 °C, 5 min | 57.6 | 87.4 | [82] |
| 6% DMF | Thai Native Chicken | 25 | BHSV | NI | 0.5 mL straw | 11 cm above the LN$_2$ for 12 min, 3 cm above the LN$_2$ for 5 min | 5 °C, 5 min | 64.3 | 91.2 | [60] |
| 6% DMF | Thai Native Chicken | 52–104 | Schramm | NI | 0.5 mL straw | 11 cm above the LN$_2$ for 12 min, 3 cm above the LN for 5 min | 5 °C, 5 min | 58.2 | 91.9 | [83] |
| 6% DMF | Thai Native Chicken | 40–63 | BHSV | Cysteamine Ergothioneine Serine | 0.5 mL straw | 11 cm above the LN for 12 min, 3 cm above the LN$_2$ for 5 min | 5 °C, 5 min | 60.1 57.6 62.7 | 69.9 66.8 90.9 | [84] |
| 6% DMF | Thai Native Chicken | NI | BHSV-Based Sasaki TNC | NI | 0.5 mL straw | 11 cm above the LN$_2$ for 12 min, 3 cm above the LN$_2$ for 5 min | 5 °C, 5 min | 68.8 68.5 66.3 | 73.4 77.3 90.3 | [61] |
| NI | Ross | 32 | Lake | Mito-TEMPO | 0.25 mL straw | 5 cm above the LN$_2$ for 12 min | 37 °C, 30 s | 60.2 | 65.3 | [85] |

NI: not informed; LN$_2$: liquid nitrogen. Lake buffer was used as the basic medium, which was composed of 0.4 g/50 mL D-fructose, 0.15 g/50 mL polyvinylpyrrolidone, 0.96 g/50 mL sodium glutamate, 0.25 g/50 mL potassium acetate, 0.035 g/50 mL magnesium acetate, 0.187 g/50 mL glycine [22].

**Table 2.** The common diluents configuration.

| Composition (g/100 mL) | Lake and Ravie | BHSV | Beltsville | EK |
|---|---|---|---|---|
| Sodium glutamate/g | 1.92 | 2.85 | 0.86 | 1.40 |
| Potassium acetate/g | 0.50 | 0.50 | 0.06 | 0.14 |
| Glucose/g | 0.80 | 0.50 | | 0.70 |
| Fructose/g | | | 0.50 | |
| Protamine sulphate/g | | | | 0.02 |
| Sodium acetate/g | | | 1.43 | |
| Magnesium acetate/g | 0.08 | 0.07 | 0.03 | |
| Polyvinylpyrrolidone/g | 0.30 | | | 0.10 |
| Dipotassium hydrogen phosphate/g | | | 1.27 | |
| Potassium dihydrogen phosphate/g | | | 0.006 | 0.21 |
| Disodium hydrogen phosphate | | | | 0.98 |
| Myo inositol/g | | 0.25 | | |
| Trihydroxyamino ethane sulfonic acid (TES)/g | | | 0.19 | |
| H$_2$O/mL | 100 | 100 | 100 | 100 |
| Osmolality/(mOsm/kg) | 340 | 380 | 330 | 385 |
| pH | 7.0 | 7.15 | 7.5 | 7.8 |
| Reference | [27] | [28] | [29] | [30] |

### 3.3. Equilibration Time

Sperm metabolism must be decreased for in vitro storage, which is practically achieved by lowering semen temperature [86]. Notably, semen temperature must be decreased gradually to prevent harmful cold shock effects [17]. A 2–4 °C equilibration is usually applied to semen to add the CPAs with the lowest possible interaction with sperm metabolism before freezing and to allow a sort of adaptation of sperm membranes and physiology that may be beneficial for post-thaw sperm survival and function. Thus, secondary equilibrium is a critical period when sperm structures are protected and ready for freezing [87]. The optimal equilibrium time depends on CPAs and experimental conditions. The procedure that diluted semen samples with DMA, loaded into 0.25-mL French straws, and equilibrated for 10 min was deemed appropriate [38]. Some authors proposed 10 min to be the optimum equilibration time before freezing turkey semen when using DMSO as the CPA [88], whereas others reported that no differences within the range of 10 to 90 min when using DMSO and EG [89]. Indeed, an equilibration time as short as 2 min has been recommended for DMA [90]; thus, the equilibration time may be shortened when using high DMA concentrations that appear to be very toxic to spermatozoa. Moreover, the choice of shorter equilibration time without deleterious effects on sperm allows the simplification of the freezing process.

### 3.4. Packaging Types

Different forms of semen packaging, such as pellets, straws, and ampoules have been used for semen cryopreservation in poultry. Glass ampoule was the first type of container introduced for chicken semen storage [86], but now it is no longer used due to it being potentially explosive. Then, plastic straws and pellets (semen drops) were developed [90] and are the main forms of frozen semen packaging at present (Table 2). Pellets are a rapid freezing method requiring less equipment, but increasing freezing homogeneity [91]. However, compared with the straw, pellet is not convenient for labeling and easy to be contaminated when removed from the freezer for manipulation [92]. Straw is suitable for rapid freezing, uniform temperature, standard dose, distinct mark, convenient thawing etc. Different CPAs with appropriate packaging types would obtain better fertility [91,93]. The French poultry semen-freezing bank mainly adopts DMA pellets/straw and glycerol straws for semen freezing [40,92]. A study on boar semen showed that the cryopreservation effect is better with 0.5 mL straw size [94]. The greater the surface-area-to-volume ratio of the French mini-straw, the better it facilitates efficient cooling and the greater the sperm cryo-survival and the post-thaw quality of the cryopreserved sperm [95]. Another study also found that 0.25 mL straws-packaged ram semen resulted in a higher lambing rate than 0.5 mL straws [96]. However, in the authors' study of chickens, straw sizes had no significant effect on the post-thaw sperm quality [53]. Therefore, to ensure the convenience of artificial insemination and semen storage, it is recommended to use 0.25 mL straws in practice due to the small volume of semen in each individual rooster.

### 3.5. Freezing and Thawing Rates

Many freezing methods with different CPAs, different types of sperm packaging and with slow and rapid freezing procedures have been studied [45]. The cooling rate is suspected to be the major factor for the reduced survival in conventional cryopreservation methods in poultry species [9]. Freezing rates can be conducted by a programmable freezer or by layering semen straws in different distances from the vapor above the surface of liquid nitrogen. A study showed that there was no difference in the integrity of the sperm when samples were frozen by a fast-freezing technique or by a slow controlled freezing method, neither in liquid nitrogen nor in vapor-phase nitrogen [97]. However, an automatic programmable freezer can make stepwise cooling more accurate, allowing sperm to gradually adapt to the low temperature in order to prevent damage [98]. Cell freezing should take place as quickly as possible to avoid the detrimental effects of CPAs; nevertheless, it should be slow enough for cells to be dehydrated [99]. The results showed

that the rapid freezing was suitable for freezing chicken semen [100]. However, the cooling rate under different conditions needs to be explored, while the appropriate cooling rate is selected according to CPAs and concentrations. Rapid freezing is usually employed with DMA [56], whereas slow freezing rates are usually used with DMSO and glycerol [101]. When using DMF and EG, the recommended freezing rates are −15 and −1 °C/min, respectively [55].

The thawing rate and temperature may affect sperm motility and fertility, and the most suitable thawing temperature is 50~60 °C [102]. Other studies have included thawing at 5 °C for 4 min [101] and at 60 °C for 25 s [90]. However, the thawing rate also depends on the CPAs used, and when freezing semen with glycerol or DMA, ideal fertility could be obtained by thawing at 5 °C or 60 °C [56]. Most of the studies summarized in Table 1 are thawed at 4–5 °C for 3 min, 37 °C for 30 s, or 60 °C for 6 s.

## 4. Freezing Damage and Freezability Mechanism

The freezing process reduced the viability and survival of thawed sperm, and it may be due to the disruption of the functional integrity of the sperm acrosome, plasma membrane, DNA and mitochondria during the freezing [103,104]. Our lab studied chicken sperm freezing and thawing [53], and we found that the freezing process caused structural damage to most spermatozoa, and the damage was greater to the mitochondria, mid-piece, and perforatorium than other parts (Figure 3) (unpublished). The fresh sperm mitochondrial membrane was continuous and the mitochondrial matrix was dense and homogeneous (Figure 3A,E). However, the density of the matrix was reduced, and the mitochondria appeared slightly swollen after freezing–thawing (Figure 3B,F). The fresh sperm perforatorium was cone-shaped and homogeneous in its contents. The boundary was clearly seen, and the perforatorium was slightly condensed (Figure 3C,G). However, a larger space appeared between the acrosome and perforatorium after thawing (Figure 3D,H).

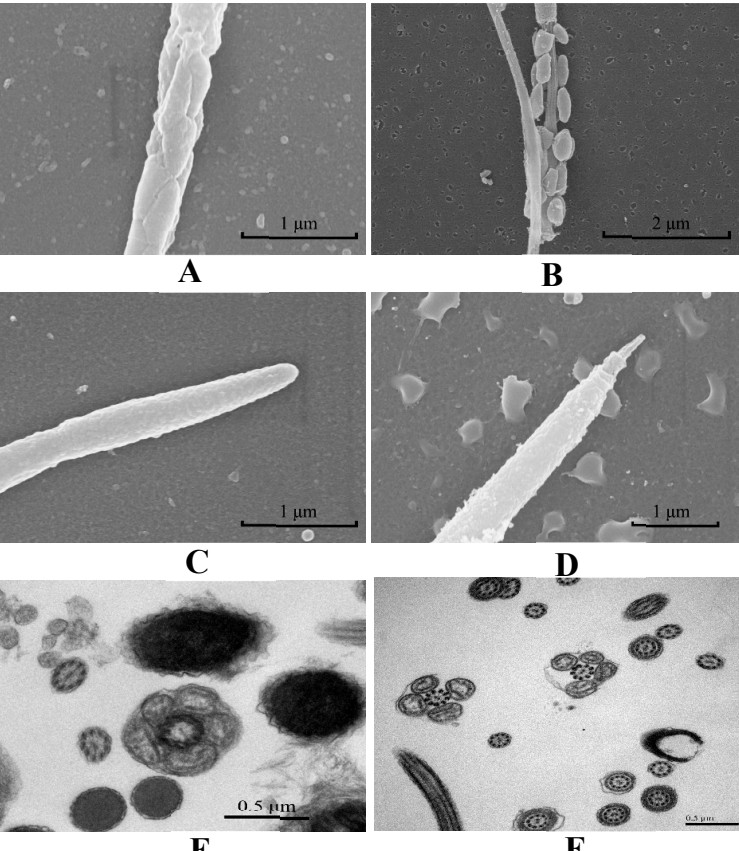

**Figure 3.** *Cont.*

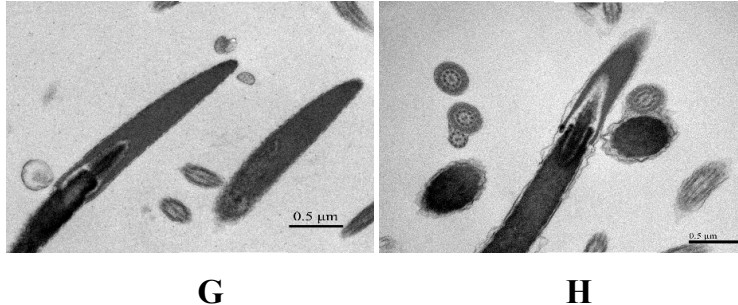

**Figure 3.** Scanning and transmission electron microscopy of fresh chicken sperm and post-thaw chicken sperm (scanning electron microscope, Hitachi SU8010, Japan; transmission electron microscope, Hitachi H-7500, Japan). (**A**) Fresh sperm mitochondria (scale bar = 1 μm at 40,000× magnification); (**B**) Post-thaw sperm mitochondria (scale bar = 2 μm at 22,000× magnification); (**C**) Fresh sperm perforatorium (scale bar = 1 μm at 40,000× magnification); (**D**) Post-thaw sperm perforatorium (scale bar = 1 μm at 30,000× magnification); (**E**) Fresh sperm mitochondria (scale bar = 0.5 μm at 70,000× magnification); (**F**) Post-thaw sperm mitochondria (scale bar = 0.5 μm at 40,000× magnification); (**G**) Fresh sperm perforatorium (scale bar = 0.5 μm at 40,000× magnification); (**H**) Post-thaw sperm perforatorium (scale bar = 0.5 μm at 25,000× magnification).

These damages may be attributed to ice crystallization damage, oxidative stress, heat shock, and osmotic shock [105]. Poultry sperm has less cytoplasm and mitochondria and a larger amount of polyunsaturated fatty acids in the plasma membrane than some mammalian species, such as human and bovine [6]. These characteristics are believed to make these cells more vulnerable to damage in the freezing process. During the freezing process, the chance of damage to the sperm plasma membrane greatly increases due to the crystallization of water [50]. The imbalance between cellular antioxidant defense systems and reactive oxygen species (ROS) production during freezing leads to oxidative stress, which some reports claim is the main cause of the sperm-freezing damage [106]. The ROS can induce detrimental changes during the process of cryopreservation of rooster semen [107,108]. The increased ROS together with reduced antioxidants can result in extensive changes in the plasma membrane and its function [109]. Changes in the sperm mitochondrial membrane potential were also associated with an increase in ROS, which altered the activity of enzymes associated with ATP production in mitochondria and further mitochondria DNA damage [110,111].

Under normal circumstances, sperm and seminal plasma have antioxidant systems that remove the ROS and prevent internal cell damage. There are two types of antioxidants: enzymatic antioxidants such as glutathione peroxidase (GPx), glutathione reductase (GR), superoxide dismutase (SOD), and catalase (CAT). Figure 4 provides the reactions catalyzed by GPx, SOD, CAT and GR to deal with ROS. Non-enzymatic antioxidants, such as vitamin E, selenium, cysteine, L-carnitine, taurine, resveratrol, and hyaluronic acid [112]. The addition of antioxidants such as vitamin E, selenium, cysteine, superoxide dismutase, L-carnitine, taurine, resveratrol, and hyaluronic acid to the freezing extenders counteracts such negative effects during the cryopreservation of rooster semen (Table 2). In recent years, many researches have focused on the application of the nanostructured lipid carrier (NLC). The cryopreservation of rooster sperm with 15 mM quercetin-loaded NLC group improved sperm quality parameters, and it may be the high potential antioxidant for the improvement of rooster pos-thaw sperm fertility performance [6]. Furthermore, when there was supplementation of the extender with 40 μM resveratrol and resveratrol-loaded NLC, there was also a positive effect on the fertility of post-thaw rooster sperm [113]. It is concluded that 15% egg yolk can be used in the cryopreservation protocol of Indian red jungle fowl sperm, which results in higher motility, plasma membrane integrity, viability, acrosome integrity and fertility [69]. Egg yolk and soybean lecithin are successfully used as cryopreservation mediums for sperm cryopreservation in a variety of species. A study concluded that the extender containing 20% egg yolk plasma and 1% soybean lecithin

resulted in higher quality of frozen–thawed sperm, and it may be a good alternative to conventional extenders that contain the whole egg yolk [78]. The study first studied the influence of soy lecithin nanoparticles on the cryopreservation of rooster semen, and found that soy lecithin nanoparticles dosage of 1.0% in the semen extender had a positive influence on the post-thaw quality in roosters, improving various sperm motion parameters, and reducing the oxidative stress during the cryopreservation process [114]. In a word, new semen freezing additives are constantly being developed, which should be further explored in combination with new technologies to improve semen-freezing efficiency.

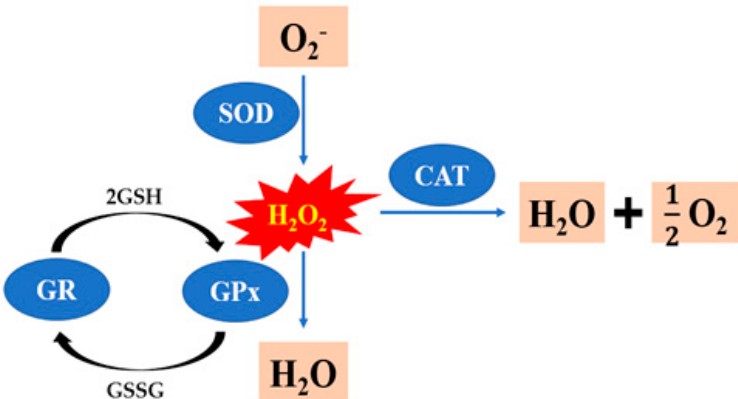

**Figure 4.** The reactions catalyzed by glutathione peroxidase (GPx), glutathione reductase (GR), superoxide dismutase (SOD), and catalase (CAT) to deal with ROS ($H_2O_2$).

The success and efficiency of the cryopreservation of semen from birds or mammals differ among species, breeds and lines. These differences in part relate to intrinsic sperm "freezability". Bulls, rams, horses, and boars have been reported to be more sensitive to thermal shock than humans, rabbits, cats, and dogs [115]. Studies have shown that there are differences in the freezing resistance of sperm from individual boars [116], even from the same ejaculation [117]. The freezability of poultry spermatozoa is low compared with that of a number of (mammalian) species [19]. Membranes are associated with the resistance of sperm to cooling, which is influenced by the chemical composition or temperature changes of the extender, which may be used to predict the fertility of frozen sperm [114]. Different varieties of poultry sperm have different levels of lipid composition on the surface of the plasma membrane, which may affect its quality to withstand against various storage conditions such as cooling and freezing [83]. The contents of fructose, MDA and SOD, in spermatozoa were closely related to the freezability of spermatozoa. It was reported that MDA in the seminal plasma significantly increased after freezing and thawing, while SOD activity in seminal plasma significantly decreased [118]. In addition, some results formally test and support the hypothesis that inter-individual differences in sperm freezability are genetically inherited rather than being random [15]. Recent developments in the various OMICS technologies lead to a better understanding of semen molecular mechanisms involved in sperm fertility. It has become popular to study the effects of chicken sperm cryopreservation using newly emerging various OMICS technologies. The identification and validation of OMICS biomarkers, such as some genes, proteins, and metabolites, related to seminal plasma and sperm greatly impact the improvement of the reproductive performance of roosters [119]. Proteomics studies of sperm decipher and identify the biomarkers of sperm freezability and fertility [120–123]. These studies revealed that the most impacted proteins during the freezing–thawing process were involved in energy metabolism, hydrolase activity, signal transduction, and sperm motility. Some stress-related genes were also observed in sperm transcriptomes, such as *CIRBP, RHOA, HSP70,* and *HSP90* [124]. *ODF2, HSP90AA1, AKAP3, AKAP4, VDAC2, TP1,* and *ACRBP* were associated with good freezability of sperm [125]. High *AKAP4* or *AKAP3* expression in frozen–thawed sperm was associated with premature capacitation [126]. RNA sequencing

(RNA Seq) has been used to evaluate RNA and quantity in post-thaw sperm, and some mRNA, microRNA (miRNA), and small non-coding RNA may be used as biomarkers of male reproductive performance [127]. Some metabolites, such as 2-oxoglutarate and fructose, have been identified as potential biomarkers for the quality and fertility of frozen bull sperm [128]. The cryopreservation of sperm leads to a significant decrease in DNA methylation, H3K9 acetylation, and H3K4 methylation [129]. The mechanisms underlying differences in sensitivity to sperm freezing in different species are complex as there are potential confounding factors to be considered.

## 5. Conclusions

In conclusion, sperm cryopreservation is a suitable method to preserve and expand invaluable poultry genetic resources. The current research field is rich and has made great progress. This significant research topic deserves further effort focusing on the exploration of alternative non-toxic CPAs and refines their protective effects. Modern omics technology should be used to further explore and reveal the mechanism of sperm freezability to generate new strategies to improve chicken sperm cryopreservation efficiency.

**Author Contributions:** Conceptualization, J.C., Y.L. and Y.Z.; data curation, Y.Z., Y.L., Y.S. and T.M.; Resources, J.C. and Y.S.; Writing-original draft, Y.Z. and Y.L.; Writing-review and editing, Y.Z., Y.S., Y.L., G.M.K.M., T.M. and J.C. All authors contributed to the interpretation of the article. All authors have read and agreed to the published version of the manuscript.

**Funding:** National Natural Science Foundation of China and The Egyptian Academy of Scientific Research and Technology (grant number 31961143028), National Key Research and Development Program of China (grant number 2021YFD1200305), China Agriculture Research Systems (grant number CARS-40), and National Germplasm Bank of Domestic Animals (2021–2022).

**Institutional Review Board Statement:** Not applicable.

**Informed Consent Statement:** Not applicable.

**Data Availability Statement:** Not applicable.

**Conflicts of Interest:** The authors declare no conflict of interest.

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
