# Peer review of "Chicken Sperm Cryopreservation: Review of Techniques, Freezing Damage, and Freezability Mechanisms"

_agriculture, doi:10.3390/agriculture13020445_

Round 1

Reviewer 1 Report

In this manuscript, the key points of the chicken sperm cryopreservation technology, including the semen dilution, cryoprotectants, equilibration time, packaging types, freezing and thawing rates balance time, and the mechanism of freezing damage were summarized and has certain reference value for the research of poultry semen cryopreservation technology. However, before the manuscript is officially published, the following problems still need to be modified:

1.   In the section of “3.1. Semen dilution”

The diluent configuration and dilution rate are important components of semen dilution. Some scholars have carried out a lot of relevant research, it is suggested to add this part.

2.   In the section of “3.2. Cryoprotectants”:

In the 2rd paragraph(L139), Please supplement the specific impact mode or mechanism.

3.   In the section of “3.3. Equilibration time”:

Please supplement the experimental data source of L206-L209.

4.   In the section of “3.5. Freezing and thawing rates”:

The authors refer that “A study showed that there was no difference in sperm quality when samples were frozen by fast-freezing technique or by slow, controlled freezing method, either in liquid nitrogen or vapor-phase nitrogen [94].” But the reference refers to “the integrity of the sperm”, not the “sperm quality”, please check the correctness of the referenced articles.

Reviewer 2 Report

Dear authors, greetings!

The manuscript "Chicken Sperm Cryopreservation: Review of Techniques, Freezing Damage, and Freezability Mechanisms" lacks a clear approach on the real importance of this type of technology regarding chicken and fail to address recent technologies such as omics.

In introduction, it is necessary to clearly demonstrate why it is important to preserve chickens’ sperm: there is an interest regarding market? A specific bird species is being preserved using this technique? How much money it involves worldwide? Why chicken is being addressed in the review and not fish or any mammal, for example? Why is this review relevant to aviculture?

Regarding Figures: for figure 1, I could not find reference to it inside the manuscript. Regarding its caption, it should be autoexplicative, so it is necessary to mention what type of microscope was used to acquire the image and what type of microscopy it is.  When it comes to figure 3, it also lacks, in caption, information regarding microscope, and it also lacks scale bar.

Regarding manuscript's structure: It fails to address some important issues. Regarding item 4. I believe it could benefit reader’s comprehension to add a Figure containing the reactions catalyzed by GPx, SOD, catalase and GR to deal with ROS. In item 3.2 I believe it could also be dedicated attention to commercial formulation aiming cryoprotection such as Poultry media® and Raptac®. What are the main commercial formulations available? Are they effective? Example of article addressing this aspect: https://www.mdpi.com/2076-2615/12/20/2886/pdf.

It would be an interesting addition to the paper, also, a new section: 5. Methods to evaluate chicken sperm viability. This topic could emphasize technique and parameters that allow verifying, after cryopreservation, the viability of sperm.

To be aligned with recent trends, the manuscript needs to address signaling as it is of interest when it comes to the opportunity to access omics data regarding cells of interest under specific conditions. Which are the main genes, transcripts and proteins already identified as related to cryopreservation success, and what are their functions?  This article is an example of reference to be used to address this: https://www.frontiersin.org/articles/10.3389/fvets.2021.609180/full.

English is fine; only minor spell check is required to remove typos and adjust some prepositions. 

Reviewer 3 Report

agriculture-2058427 – Review: Zong et al. Chicken sperm cryo

 General remark: The manuscript is well written, but would still need to be corrected by a native English speaker. For example: Line 15-17: ...many studies have been performed to .... and achieve great progress. You probably do not mean that the many studies were performed to achieve great progress. I guess this sentence contains two parts: ....experiments have been performed ... and  .... which led to great progress. So, perhaps better use past tense .... achieved great progress. Second example: Lines 17-18. I guess that the objective was not to adapt subspecies and breeds, but rather to adapt methods for different subspecies and breeds.  These are just a few examples. The whole manuscript should be checked for correct English grammar and style.

Line 37: Schaffner et al. 1941 repeated experiments similar to the ones done by earlier authors, to which Schaffner et al. refer (including for chicken: Nelson 1939). So it is not true that “ ... research on the cryopreservation of chicken semen began in 1941”. All these studies did not succeed in preserving fertility of the sperm, so, authors may wish to also mention the first study that did result in successful preservation of fertility of chicken sperm (Polge et al., 1949).

Change post-thawed to post-thaw throughout the manuscript

Line 59-60. More susceptible than what? It is difficult to make a fair comparison with other species on the susceptibility of sperm cells. We can say though with some confidence that successful cryopreservation of fertility of semen appears to be easier, more successful, for some species such as bovine, than for chicken. But, see also Holt 2000, who argues that chicken (and bovine) are probably better freezable than rodent and porcine.

Line 65-66 Unclear. ... surrounded by mitochondria arranged into outer mitochondria ....

Line 66-68. The size and shape do not affect the membrane permeability, but the shape and surface to volume ratio affect the rate of equilibration of water. Change to something like ...Cell volume and shape affect the membrane surface area to cell water volume ratio, and therewith their ability to respond to osmotic changes.

Line 70-73. I can see that the very long and slender tail of a chicken sperm cell can make it vulnerable. I do not find this in reference [17] (Holt, 2000), though.

Line 77-78. Change to: .... oxidative stress, which may lead to reduced ....

Line 79: Change to bis-allylic methylene groups of plasma membrane phospholipids

Line 80. Check this reference. These authors say this, but refer to an earlier paper. But in that earlier paper there is no mention of such phospholipid peroxidation.

Line 101. This is incorrect. Semen from commercial chicken breeds may have 4000 or even higher sperm cells per ml. In fact, 200-700 million /ml would be low compared with some other species, such as in cattle.

Line 103 and 105: Change osmolarity to osmolality. The osmotic pressure as measured with an osmometer is expressed as Osm./kg water and is called osmolality.

Line 133: Extender simply means a medium to extend, or dilute semen. Reserve the words extender and diluent for a medium for fresh semen and use the word freezing medium for media used for freezing.

Line 145. Lake extender. Which one. Please explain. There are many publications from lake and there are a number of diluents (extenders) and freezing media, all with different recipes. Also in Table 1 it is not sufficient to just say “Lake” for the extender used. It should be specified also in Table 1.

Line 145: .... from 1300 mOsm/kg ... That depends on the used glycerol concentration. So refer to the concentration used in reference [46].

Lines 154-157. Many earlier publications show successful freezing with good fertility using straw freezing with DMA.

Lines 171-172. “ ... EG has lower molecular weight, lower toxicity, and higher permeability ....”. If you write a review paper, you should be critical about things you copy from other papers. Read the original papers and critically evaluate what you read. There is no study that measured permeability coefficient for EG for chicken sperm. Your reference [59] never measured permeability coefficients for EG in any cell. They refer to an earlier paper by Massip who also didn’t measure it, and Massip referred to earlier Japanese studies who also didn’t measure it. The latter studies were about embryo freezing. These authors actually showed that permeation of EG isn’t even required for its protection of embryos! Having said that, there are indeed (other) papers that do show that permeability in embryos is higher for EG than for glycerol. This may be relevant for the (large) volume excursions of embryos during addition and removal of CPA. But, the permeability differences between glycerol and EG, DMA,DMSO, etc, play no role in the suitability as a cryoprotectant for sperm cells. For sperm cells, the volume excursions during addition of CPA and during post-thaw removal, are extremely shallow and extremely short. That means that for sperm cells, the osmotic stress of addition and removal of CPA is negligible. And during freezing and thawing, the equilibration of water (rather than of CPA) is the critical component. There is no reason to believe that the permeability differences between glycerol and EG, DMA,DMSO, etc, play no role in the suitability as a cryoprotectant for sperm cells.

Table 1. Specify Lake. See earlier comment.

Table 1. Should reference 78 perhaps be 87 (Tselutin et al., DMA, pellet method)?

Line 199: “ ... to ensure an equilibration of CPAs and sperm.”  The entry of CPA into perm cells is extremely fast. It requires less than 1 minute. The main purpose of a holding time after adding CPAs is not equilibration in a physical sense, but rather to allow some sort of adaptation of sperm membranes and/or physiology that may be beneficial for their post-thaw survival and function. For DMA, you may refer to Tselutin et al., 1995, who used very short times after adding DMA, or Woelders et al., 2022, who used 1 h holding.

Line 216. It seems that [88] is incorrect, as this study used straw freezing. I guess you mean [87]

Line 223: I think reference [91] did not study effect of straw size. Only effect of seminal plasma.

Line 224. Neither did reference [92]. This study used only part of the length of a ministraw to study effect of degree of dilution of the (bull) semen.

Lines 237-248: I think that the review of cooling rates should mention the resent comprehensive study by Woelders et al., 2022.

Line 250: “... the thawing rate should be consistent with the freezing ...” I no of no study that supports this. On the contrary, most reports state that highest possible warming rates are beneficial.

Line 260. Greater than what? And I do not think that these studies [100, 101] really compare structural damage of mitochondria versus other structures of the cell. Nor can this be judged from figure 3, as these pictures do not allow numerical comparison of different structures, let alone any statistical comparison. And, from which study are these nice ultrastructural pictures of figure 3? And how was the semen frozen? If this is from author’s own (unpublished) studies, please say so.

Line 269-271. Please state this less strongly: These damages may be attributed to ...

Lines 270-271. Less than which cells? If you mean compared with sperm cells of mammalian species, I am not sure this (less cytoplasm) is true.

Line 272-273. Regarding amount of polyunsaturated fatty acids in the plasma membrane please change to: These characteristics are believed to make these cells more vulnerable to damage in the freezing process ....

Line 276. Change to: .... which some reports claim to be the main cause of sperm freezing damage [103]

Line 308-309. Incorrect. Please read the paper you refer to more carefully. What these authors say is that spem cells of these species are sensitive to thermal shock. But, as we all know, that does not necessarily mean they are cryosensitive and cannot easily be frozen. It simply means that if you cool from body temperature to 4°C, (prior to freezing) the cooling rate must not be too high. For instance we all know that bull semen can very well be frozen, provided that cooling to 4°C is done gradually( e.g. <1 °C/min).

Line 310. I don’t think that this study compared freezability of sperm within an ejaculate.

Line 311-312. This is a much too strong statement. At best you can say that freezability of poultry spermatozoa is low compared with that of a number of (mammalian) species.

Line 312-313. Where in reference [112] can we read that “sperm plasma membrane fluidity is related 312 to sperm freezability”  and from which original studies was that statement taken? I think reference [112]  only showed that fertility is better when chicken sperm was frozen with soybean lecithin. No measurements of membrane fluidity there.

Line 315. There is no study that would say cholesterol on the surface of the plasma membrane affects the ability of CPA to penetrate. Unless, with penetration you mean perhaps partitioning into the membrane. If that is what you mean, please give the proper reference for that statement.

Check your reference list. Reference 78 is not on a new line.

 agriculture-2058427 – Review: Zong et al. Chicken sperm cryo

General remark: The manuscript is well written, but would still need to be corrected by a native English speaker. For example: Line 15-17: ...many studies have been performed to .... and achieve great progress. You probably do not mean that the many studies were performed to achieve great progress. I guess this sentence contains two parts: ....experiments have been performed ... and  .... which led to great progress. So, perhaps better use past tense .... achieved great progress. Second example: Lines 17-18. I guess that the objective was not to adapt subspecies and breeds, but rather to adapt methods for different subspecies and breeds.  These are just a few examples. The whole manuscript should be checked for correct English grammar and style.

Line 37: Schaffner et al. 1941 repeated experiments similar to the ones done by earlier authors, to which Schaffner et al. refer (including for chicken: Nelson 1939). So it is not true that “ ... research on the cryopreservation of chicken semen began in 1941”. All these studies did not succeed in preserving fertility of the sperm, so, authors may wish to also mention the first study that did result in successful preservation of fertility of chicken sperm (Polge et al., 1949).

Change post-thawed to post-thaw throughout the manuscript

Line 59-60. More susceptible than what? It is difficult to make a fair comparison with other species on the susceptibility of sperm cells. We can say though with some confidence that successful cryopreservation of fertility of semen appears to be easier, more successful, for some species such as bovine, than for chicken. But, see also Holt 2000, who argues that chicken (and bovine) are probably better freezable than rodent and porcine.

Line 65-66 Unclear. ... surrounded by mitochondria arranged into outer mitochondria ....

Line 66-68. The size and shape do not affect the membrane permeability, but the shape and surface to volume ratio affect the rate of equilibration of water. Change to something like ...Cell volume and shape affect the membrane surface area to cell water volume ratio, and therewith their ability to respond to osmotic changes.

Line 70-73. I can see that the very long and slender tail of a chicken sperm cell can make it vulnerable. I do not find this in reference [17] (Holt, 2000), though.

Line 77-78. Change to: .... oxidative stress, which may lead to reduced ....

Line 79: Change to bis-allylic methylene groups of plasma membrane phospholipids

Line 80. Check this reference. These authors say this, but refer to an earlier paper. But in that earlier paper there is no mention of such phospholipid peroxidation.

Line 101. This is incorrect. Semen from commercial chicken breeds may have 4000 or even higher sperm cells per ml. In fact, 200-700 million /ml would be low compared with some other species, such as in cattle.

Line 103 and 105: Change osmolarity to osmolality. The osmotic pressure as measured with an osmometer is expressed as Osm./kg water and is called osmolality.

Line 133: Extender simply means a medium to extend, or dilute semen. Reserve the words extender and diluent for a medium for fresh semen and use the word freezing medium for media used for freezing.

Line 145. Lake extender. Which one. Please explain. There are many publications from lake and there are a number of diluents (extenders) and freezing media, all with different recipes. Also in Table 1 it is not sufficient to just say “Lake” for the extender used. It should be specified also in Table 1.

Line 145: .... from 1300 mOsm/kg ... That depends on the used glycerol concentration. So refer to the concentration used in reference [46].

Lines 154-157. Many earlier publications show successful freezing with good fertility using straw freezing with DMA.

Lines 171-172. “ ... EG has lower molecular weight, lower toxicity, and higher permeability ....”. If you write a review paper, you should be critical about things you copy from other papers. Read the original papers and critically evaluate what you read. There is no study that measured permeability coefficient for EG for chicken sperm. Your reference [59] never measured permeability coefficients for EG in any cell. They refer to an earlier paper by Massip who also didn’t measure it, and Massip referred to earlier Japanese studies who also didn’t measure it. The latter studies were about embryo freezing. These authors actually showed that permeation of EG isn’t even required for its protection of embryos! Having said that, there are indeed (other) papers that do show that permeability in embryos is higher for EG than for glycerol. This may be relevant for the (large) volume excursions of embryos during addition and removal of CPA. But, the permeability differences between glycerol and EG, DMA,DMSO, etc, play no role in the suitability as a cryoprotectant for sperm cells. For sperm cells, the volume excursions during addition of CPA and during post-thaw removal, are extremely shallow and extremely short. That means that for sperm cells, the osmotic stress of addition and removal of CPA is negligible. And during freezing and thawing, the equilibration of water (rather than of CPA) is the critical component. There is no reason to believe that the permeability differences between glycerol and EG, DMA,DMSO, etc, play no role in the suitability as a cryoprotectant for sperm cells.

Table 1. Specify Lake. See earlier comment.

Table 1. Should reference 78 perhaps be 87 (Tselutin et al., DMA, pellet method)?

Line 199: “ ... to ensure an equilibration of CPAs and sperm.”  The entry of CPA into perm cells is extremely fast. It requires less than 1 minute. The main purpose of a holding time after adding CPAs is not equilibration in a physical sense, but rather to allow some sort of adaptation of sperm membranes and/or physiology that may be beneficial for their post-thaw survival and function. For DMA, you may refer to Tselutin et al., 1995, who used very short times after adding DMA, or Woelders et al., 2022, who used 1 h holding.

Line 216. It seems that [88] is incorrect, as this study used straw freezing. I guess you mean [87]

Line 223: I think reference [91] did not study effect of straw size. Only effect of seminal plasma.

Line 224. Neither did reference [92]. This study used only part of the length of a ministraw to study effect of degree of dilution of the (bull) semen.

Lines 237-248: I think that the review of cooling rates should mention the resent comprehensive study by Woelders et al., 2022.

Line 250: “... the thawing rate should be consistent with the freezing ...” I no of no study that supports this. On the contrary, most reports state that highest possible warming rates are beneficial.

Line 260. Greater than what? And I do not think that these studies [100, 101] really compare structural damage of mitochondria versus other structures of the cell. Nor can this be judged from figure 3, as these pictures do not allow numerical comparison of different structures, let alone any statistical comparison. And, from which study are these nice ultrastructural pictures of figure 3? And how was the semen frozen? If this is from author’s own (unpublished) studies, please say so.

Line 269-271. Please state this less strongly: These damages may be attributed to ...

Lines 270-271. Less than which cells? If you mean compared with sperm cells of mammalian species, I am not sure this (less cytoplasm) is true.

Line 272-273. Regarding amount of polyunsaturated fatty acids in the plasma membrane please change to: These characteristics are believed to make these cells more vulnerable to damage in the freezing process ....

Line 276. Change to: .... which some reports claim to be the main cause of sperm freezing damage [103]

Line 308-309. Incorrect. Please read the paper you refer to more carefully. What these authors say is that spem cells of these species are sensitive to thermal shock. But, as we all know, that does not necessarily mean they are cryosensitive and cannot easily be frozen. It simply means that if you cool from body temperature to 4°C, (prior to freezing) the cooling rate must not be too high. For instance we all know that bull semen can very well be frozen, provided that cooling to 4°C is done gradually( e.g. <1 °C/min).

Line 310. I don’t think that this study compared freezability of sperm within an ejaculate.

Line 311-312. This is a much too strong statement. At best you can say that freezability of poultry spermatozoa is low compared with that of a number of (mammalian) species.

Line 312-313. Where in reference [112] can we read that “sperm plasma membrane fluidity is related 312 to sperm freezability”  and from which original studies was that statement taken? I think reference [112]  only showed that fertility is better when chicken sperm was frozen with soybean lecithin. No measurements of membrane fluidity there.

Line 315. There is no study that would say cholesterol on the surface of the plasma membrane affects the ability of CPA to penetrate. Unless, with penetration you mean perhaps partitioning into the membrane. If that is what you mean, please give the proper reference for that statement.

Check your reference list. E.g. Grötter is incorrect. And Reference 78 is not on a new line.

Reviewer 4 Report

In my opinion, the review entitled “Chicken Sperm Cryopreservation: Review of Techniques, Freezing Damage, and Freezability Mechanisms” is certainly appreciable for the structure and contents.

In this present form, this manuscript is really a valuable, well-arranged review, focusing on the research outline of sperm cryopreservation technology in poultry including different processes of chicken sperm cryopreservation. The different processes of chicken sperm cryopreservation are deeply discussed, supported by several and appropriate recent references.

In order to rich the review, I suggest the authors include this titles in the review:

-          Egg yolk and soybean lecithin based extender in cryopreservation of poultry semen.

-          Using nanotechnology in counteract the cryo-damages in poultry semen cryopreservation.

Round 2

Reviewer 2 Report

Dear authors, all the aspects addressed in previous comments were successfully improved.